

# A validation standard for Area of Habitat maps for terrestrial birds and mammals

Prabhat R. Dahal[1,2] , Maria Lumbierres[1,2], Stuart H. M. Butchart[2,3], Paul F. Donald[2,3], Carlo Rondinini[1]

**Affiliations**

1- Global Mammal Assessment Program, Department of Biology and Biotechnologies, Sapienza University of Rome, Viale dell'Università 32, 00185 Rome, Italy

2- BirdLife International, David Attenborough Building, Pembroke Street, Cambridge CB2 3QZ, UK

3- Department of Zoology, University of Cambridge, Downing Street, Cambridge CB2 3EJ, UK

corresponding author: prabhatraj.dahal@uniroma1.it

**Abstract**

Area of Habitat (AOH) is a deductive model which maps the distribution of suitable habitat at suitable altitudes for a species inside its broad geographical range. AOH maps have been validated using presence-only data for small subsets of species for different taxonomic groups, but no standard validation method exists when absence data are not available. We develop a novel two-step validation protocol for AOH which includes first a model-based evaluation of model prevalence (i.e, the proportion of suitable habitat within a species' range), and second a validation using species point localities (presence-only) data. We applied the protocol to AOH maps of terrestrial birds and mammals. In the first step we built logistic regression models to predict expected model prevalence (the proportion of the range retained as AOH) as a function of each species' elevation range, mid-point of elevation range, number of habitats, realm and, for birds, seasonality. AOH maps with large difference between observed and predicted model prevalence were identified as outliers and used to identify a number of sources of systematic error which were then corrected when possible. For the corrected AOH, only 1.7% of AOH maps for birds and 2.3% of AOH maps for mammals were flagged as outliers in terms of the difference between their observed and predicted model prevalence. In the second step we calculated point prevalence, the proportion of point localities of a species falling in pixels coded as suitable in the AOH map. We used 48,336,141 point localities for 4889 bird species and 107,061 point localities for 420 mammals. Where point prevalence exceeded model prevalence, the AOH was a better reflection of species' distribution than random. We also found that 4689 out of 4889 (95.9%) AOH maps for birds, and 399 out of 420 (95.0%) AOH maps



for mammals were better than random. Possible reasons for the poor performance of a small
proportion of AOH maps are discussed.
**Introduction**
An accurate estimate of the distribution of species is central to ecological and conservation research
and action. There are three different classes of information on the distribution of species (Rondinini
and Boitani, 2006). These are 1) point localities (latitude and longitude) of individuals; 2)
geographic ranges, which are derived by mapping the extent of known point localities along with
expert knowledge; and 3) species distribution models, which use environmental and other relevant
variables associated with the species to refine geographical ranges. Species distribution models are
of two types (Stoms et al., 1992). The first are deductive models, which use expert-based
information on species' habitat use to model the suitable areas for the species. The second type are
inductive models, in which the environmental conditions at point localities where the species were
recorded are interpolated over wider areas.
Area of Habitat (AOH; also known as Extent of Suitable Habitat, ESH) is a deductive model which
maps the distribution of suitable habitat for a species inside its broad geographical range (Brooks et
al., 2019). It aims to reduce commission errors present in the range map while minimizing omission
errors. Several sets of AOH maps for different taxonomic groups at continental and global scales
have already been produced (Rondinini et al., 2005; Rondinini et al., 2006; Catullo et al., 2008;
Jenkins and Giri, 2008; Rondinini et al., 2011; Ficetola et al., 2015; Tracewski et al., 2016;
Lumbierres et al., 2021b).
Habitat models are prone to two major types of errors: omission errors occur when suitable habitat
areas for the species are wrongly mapped as being unsuitable, commission errors occur when areas
unsuitable for the species are wrongly mapped as being suitable. Quantification of these errors is
one of the key parts of the habitat modeling process and is done by validation. The omission and
commission errors could both be quantified only when independent presence and absence data on
the species are available. In such cases standard validation metrics such as True Skill Statistics
(TSS) (Allouche et al., 2006) and the Boyce Index (Boyce et al., 2002) are used. In case of AOH
maps produced for large taxonomic groups when true absence data are not available, no standard
validation method exists.
Rondinini et al. (2011) and Ficetola et al. (2015) used point localities from GBIF (Global
Biodiversity Information Facility) (www.gbif.org) to validate AOH maps for mammals and
amphibians respectively. AOH maps for South Asian mammals (Catullo et al., 2008) and African





vertebrates (Rondinini et al., 2005) were also validated using point localities. Brooks et al. (2019)
recommend using point localities for validation and inclusion of AOH maps for IUCN
(International Union for Conservation of Nature) Red List assessment. However, point localities are
often not available for many species and are biased towards certain taxonomic group and well-
studied areas.
In this paper, we developed a novel two-step validation protocol for AOH which includes: a) a
model-based evaluation of model prevalence (i.e., the proportion of a species' range that comprises
AOH), and b) a validation using species point localities (presence-only) data. We demonstrate the
use of this approach by validating a new set of AOH maps produced by Lumbierres et al. (2021b)
for all terrestrial birds and mammals. The validation method developed here is an iterative process
whereby systematic errors in the production of AOH (e.g. in the matching of habitat classes to land
cover maps) were identified using logistic regression models, then corrected where possible and a
new set of AOH maps produced. Then we employed a point validation analysis for the subset of
species for which point localities were available to assess the performance of the AOH maps.
Finally, we assessed the extent to which the subset of species for which point locality data were
available were representative of those for which no point data were available.
**2. Methods**
The new set of AOH maps ( Lumbierres et al., 2021b) was produced at a resolution of 100 m using
a novel habitat-land cover model ( Lumbierres et al., 2021a) which associated the different land
cover classes in the Copernicus land cover map (Buchhorn et al., 2019) with the Level-1 habitat
classes of the IUCN habitat classification scheme (IUCN, 2012). The IUCN habitat classification
scheme is a hierarchy of habitat classes, and each species assessed in the IUCN Red List is assigned
to one or more of these habitat classes, based on available information in the literature, unpublished
reports and expert knowledge. The habitat-land cover model (Lumbierres et al., 2021a) has the
provision of associating IUCN habitat classes to land cover classes using three different thresholds
(1, 2 and 3). Lower thresholds permit weaker associations between land cover and habitat classes.
Therefore, with threshold 1 each land cover class is associated with more habitat classes than with
threshold 3. Lumbierres et al. (2021b) produced a set of AOH maps for each of the three different
thresholds by clipping out of each species' range any cells of land cover that were not linked by the
model to the habitat class(es) to which the species was coded, then further clipping out parts of the
range falling outside the elevation range of the species.





In order to identify the best threshold among the three thresholds and to validate the set of AOH
maps with the best threshold at species level, we quantified two measures: 'model prevalence' and
'point prevalence'. Model prevalence is defined as the proportion of pixels inside the range that
were retained in the AOH. For example, if 25% of the pixels present in the original range map are
clipped out because they contain unsuitable habitat, fall outside the species' elevation range or both,
the model prevalence is 0.75. Point prevalence is defined as the proportion of point localities (or
their buffers) out of all points inside the range of a species falling inside the suitable pixels. For
example the Red-tailed Comet (*Sappho sparganurus*) had a total of 71 point localities within its
range, of which 62 fell in pixels coded as suitable in the species' AOH map, giving a point
prevalence of 62/71 = 0.88.
Because the number of habitats associated with each land cover class decreases with increasing
thresholds, model prevalence is highest for threshold 1 models and lowest for threshold 3 models.
With increasing threshold, commission errors are expected to decrease (which is the main purpose
of AOH) but omission errors might increase. Our validation protocol therefore aimed to control for
omission errors. We did this by calculating point prevalence and model prevalence across the three
thresholds and identified the set of AOH maps for which the mean model prevalence was lowest
without compromising the mean point prevalence.
The point localities for bird species were downloaded from eBird (www.ebird.org), the largest
global repository for data on point localities of birds. eBird provides a metadata file called "eBird
basic data set" (Cornell Lab of Ornithology, 2020) which is a compilation of all the validated point
localities at species level and is updated monthly. These point localities are submitted by citizen
scientists as well as experts worldwide and are checked by local experts to remove obvious
misidentifications before they are made available for download (Sullivan et al., 2009). We first
downloaded the metadata file from eBird updated in January 2020 which was then queried in R (R
Core Team, 2018) using the *auk* package (Strimas-Mackey et al., 2018), as recommended by eBird,
to extract the point localities at species level. The taxonomy of Birdlife International (BirdLife
International and Handbook of the Birds of the World, 2020), which is that followed by the IUCN,
was matched with eBird's taxonomy and point localities of only those species common to both were
queried and extracted from the metadata. Of the 10,813 species listed in Birdlife International's list
for which AOH maps were produced, 9628 species matched by name. Of these 9628 species, 8998
species shared the same taxonomic concept and for 730 species the scientific names matched but
the taxonomic concept did not.
To ensure that only high-accuracy points were used for the validation, we selected the stationary
points from eBird's metadata. The stationary points are those that have coordinate uncertainty of





less than 30 m. We then applied a temporal filter of 2019-2020 because the point localities from
2005-2018 were used to calibrate the habitat-land cover model in Lumbierres et al. (2021a). This
ensured there was no overlap between the calibration and validation data. The points were further
filtered by the range polygon of the species provided by the IUCN Red List website (IUCN, 2020)
to remove the small number of points falling outside the range (many of them likely to be
misidentifications). Since the AOH maps in question only include a certain combination of
presence, origin and seasonality of the range, we used the same combination to filter the point
localities. This ensured that we only included points which fell inside the boundaries of the selected
range maps. We also made sure that only one point locality was allowed per pixel of the AOH map
to avoid clustering of points. Finally, we excluded species which had fewer than 10 point localities
after all the filters were applied. A total of 4889 bird species had 4,836,141 point localities after
filtering. For mammals, point localities were downloaded from GBIF (Cold Spring Harbor
Laboratory, 2021) following the taxonomy of Global Mammal Assessment (which is followed by
IUCN) with same temporal and spatial filters as with birds except the filter of coordinate
uncertainty which was set to 300 m for mammals. This was done because far too many mammal
species would be excluded in the validation if we only considered point localities with coordinate
uncertainty of less than 30 m. The *rgbif* package (Chamberlain et al., 2021) in R was used to
download the points for mammals. A total of 107,061 point localities for 420 species were available
for mammals after applying all the filters.
A buffer of 300 m was applied around all the point localities to account for the positional
uncertainty of the points and for the fact that the location usually records that of the observer at the
time of observation and not the focal animal, following Jung et al. (2020). The buffers of point
localities were then overlaid on top of the AOH maps across all three thresholds at species level and
if at least one pixel coded to suitable habitat was found inside the buffer, the pixel was considered to
be validated at that point locality. The count of validated pixels was used to calculate point
prevalence at species level across all three thresholds.
We identified the threshold that produced a set of AOH maps for which the mean model prevalence
was lowest without detriment to the mean point prevalence.
We then employed a two-step approach to validate the set of AOH maps with the optimal threshold.
In the first step, we identified potential systematic errors in the AOH maps using a modeling
approach that aimed to identify species whose model prevalence was larger or smaller than
expected, given the characteristics of the species concerned. In the second step, we validated the
AOH maps using point localities following Rondinini et al. (2011).



2.1 A modeling approach to identify outliers
We used logistic generalized linear models to predict model prevalence of the set of AOH maps
produced using the optimal threshold as a function of a number of independent variables, and
identified outliers whose observed model prevalence was significantly higher or lower than
predicted by the model. Outliers were then examined to identify systematic errors in, for example,
the way habitats were coded to land cover classes in the production of the AOH maps, and to
identify species that might be coded to the wrong habitats or elevation limits. For example, if a
species' range includes a high proportion of a particular land cover type not associated with the
suitable habitats of the species in the land cover-habitat association table (Lumbierres et al., 2021b),
or if errors in coding species to elevation limits mean that most of the range is outside the species'
stated limits, the model prevalence would be lower than predicted by the model.
The predictors fitted to the logistic models included: elevation range of the species (upper elevation
limit minus lower elevation limit), mid-point of the elevation range, number of habitats to which the
species is coded against in the IUCN Red List, seasonality of species (breeding and non-breeding
ranges in case of migratory birds) and the geographical realm of the species. In case of migratory
birds, Lumbierres et al. (2021b) has three different classes ( resident, breeding and non- breeding
seasonalities) of AOH maps based on seasonality of the species. We merged resident seasonality to
breeding and non breeding seasonalities to have AOH maps with only two seasonalities ( breeding
and non-breeding). The dependent variable was the model prevalence of the AOH maps. Data from
a total of 10475 AOH maps for 9163 bird species (including for some species with separate
breeding and non-breeding ranges) and 2758 AOH maps for 2758 mammal species were used to
build logistic regression models for birds and mammals separately using the *lme4* (Bates et al.,
2015) package in R . Data on elevation were lacking for many mammal and bird species which is
the reason why not all species could be included in the logistic model. After testing taxonomic
genus, family and order as random effects in the model to control the non-independence of closely
related taxa, family was selected for fitting as the residual variance was lowest for the models with
family as the random effect for both birds and mammals. The predictive power of the model was
assessed by calculating marginal $R^2$ and conditional $R^2$ using the *insight* (Lüdecke et al., 2019)
package in R. The marginal $R^2$ expresses how much of the variation in data is explained by the fixed
effects and conditional $R^2$ tells how much of the variation in data is explained by both fixed and
random effects.
The Tukey fences outlier detection test (Wilcox, 2017) was used to identify outliers based on the
difference between the estimated and observed values of model prevalence. This test uses the





interquartile ranges to estimate the outliers in a data-set. The outlier test identified mild lower and
upper threshold values for the difference between estimated and observed values.
*Mild upper threshold = (interquartile range \* 1.5) + upper quartile*
*Mild lower threshold  = lower quartile - ( interquartile range \* 1.5)*
The AOH maps identified as mild upper outliers have an observed model prevalence much larger
than their predicted model prevalence, whereas maps identified as mild lower outliers have an
observed model prevalence much smaller than their predicted model prevalence.
In order to investigate the sources of errors in the outliers, we produced two more sets of AOH
maps for the outliers. One set included AOH maps which were produced by clipping the range of
the species by the altitudinal range only (AOH $_{\text{Elevation only}}$). Similarly, the other set included AOH
maps which were derived by clipping the range with only suitable habitat of the species (AOH $_{\text{Habitat}}$
$_{\text{only}}$). If the model prevalence of an outlier was equal or nearly equal to the model prevalence of its
AOH $_{\text{Elevation only}}$, then we concluded that the under-representation of model prevalence could be
attributed to errors in elevation range of the species. If the model prevalence of an outlier was equal
or nearly equal to the model prevalence of AOH $_{\text{Habitat only}}$, then the source of error could be attributed
to the mapping of the habitats inside the range using the habitat-land cover crosswalk (Lumbierres
et al., 2021a) or to errors in the species' habitat coding. Furthermore, in some of the outliers the
under-representation could result from inclusion of large proportion of habitats which were
unsuitable for the species but were inside the range map of the species. Outliers do not necessarily
represent errors in AOH, as species might legitimately have very high or low model prevalence, but
by identifying suites of outliers sharing common characteristics we were able to identify and correct
a number of systematic errors in AOH production. The models also allowed us to identify species
whose AOH maps might be unreliable and whose habitat and elevation coding needs to be checked.
2.2 Point validation of AOH maps of terrestrial birds and mammals
We validated 4889 bird and 420 mammal species' AOH maps using the filtered point localities. The
point validation was done by comparing the model and point prevalence at species level. If the point
prevalence exceeded model prevalence at species level, the AOH maps performed better than
random, otherwise they were no better than random. We also calculated the percentage of suitable





habitat pixels inside the buffers to ensure that the validation success wasn't due to a one off pixel
falling inside the 300 m buffer.
One of the major issues with citizen science data is that there is often a non-representative spread of
data across species. It is therefore possible that the species included in the point validation analysis
are not representative, in terms of the ratio between point prevalence and model prevalence, of the
species not included. We assessed how representative the validation sample size was by comparing
the representation of variables such as family, order, genus, realm, elevation range, mid-point of the
elevation range, range size and extinction risk categories for birds and mammals between species
with and without point data. The point validation was done in R and GRASS (GRASS Development
Team, 2017).
**3. Results**
After comparing point and model prevalence of 4889 birds and 420 mammal species across all the
three thresholds, we selected the set of AOH maps derived by using threshold 3 in the habitat-land
cover model. At threshold 3, the mean model prevalence decreased as compared to thresholds 1 and
2 with much lower change in the mean point prevalence (Table 1 and 2) for both birds and
mammals.

|  | Threshold 1 | Threshold 2 | Threshold 3 |
|---|---|---|---|
| Mean model prevalence | 0.81 ± 0.21 SD | 0.77 ± 0.23 SD | 0.65 ± 0.25 SD |
| Mean point prevalence | 0.95 ± 0.14 SD | 0.94 ± 0.14 SD | 0.90 ± 0.17 SD |

**Table 1:** Mean model and point prevalence for AOH maps with standard deviation of 4889 bird
species across 3 different thresholds.

|  | Threshold 1 | Threshold 2 | Threshold 3 |
|---|---|---|---|
| Mean model prevalence | 0.87 ± 0.21 SD | 0.83 ± 0.22 SD | 0.73 ± 0.24 SD |
| Mean point prevalence | 0.95 ± 0.14 SD | 0.95 ± 0.15 SD | 0.93 ± 0.17 SD |

**Table 2:** Mean model and point prevalence for AOH maps with standard deviation of 420 mammal
species across 3 different thresholds.
We also assessed the relative contribution of elevation range, habitat, and both in reducing the range



to AOH. For both birds and mammals, most of the pixels removed from the range were because
either the habitat or the elevation were unsuitable, with a relatively small proportion being removed
because both were unsuitable (Figs. 1,2). The proportion of the range that was clipped out on the
basis of having unsuitable habitat at suitable elevations increased as model prevalence decreased,
whereas there was little change across the same axis in the proportion of the range that was
excluded on the basis of having suitable habitat at unsuitable elevations (Figs. 1,2). The number of
both bird and mammal species peaked at model prevalence of 95-100% and gradually decreased as
the model prevalence decreased.

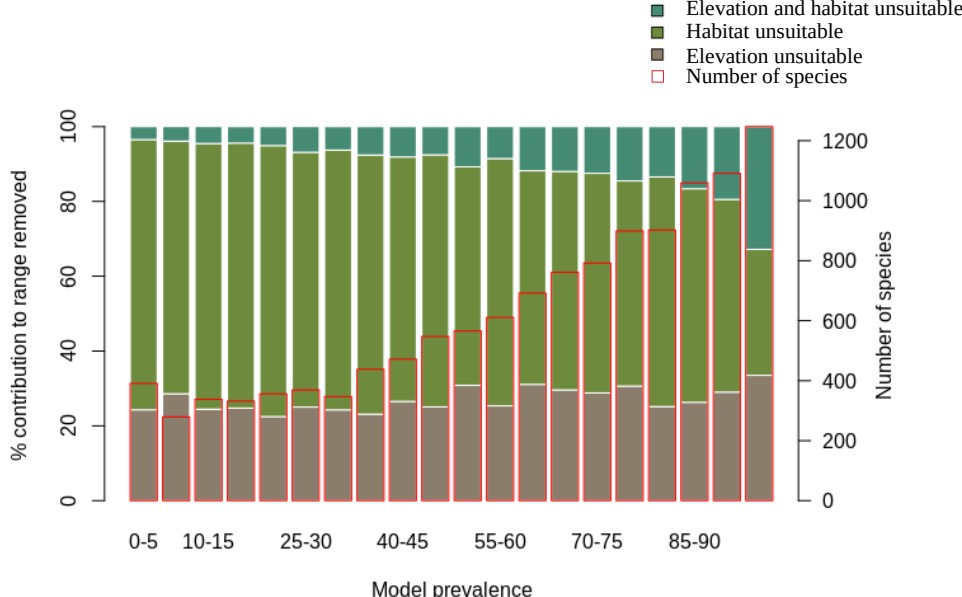

**Figure 1:** Percentage contribution of elevation range, habitat and both in clipping the IUCN range
to produce AOH maps for birds. Each bar represents a 5% bin of model prevalence, divided to show
how much of the range was clipped out due to unsuitable habitat at suitable elevations ("Habitat
unsuitable"), by suitable habitat at unsuitable elevations ("Elevation unsuitable") and by unsuitable
habitat at unsuitable elevations ("Elevation and habitat unsuitable"). The red blocks correspond to
the second *y*-axis and show the number of species falling into each 5 % bin of model prevalence.



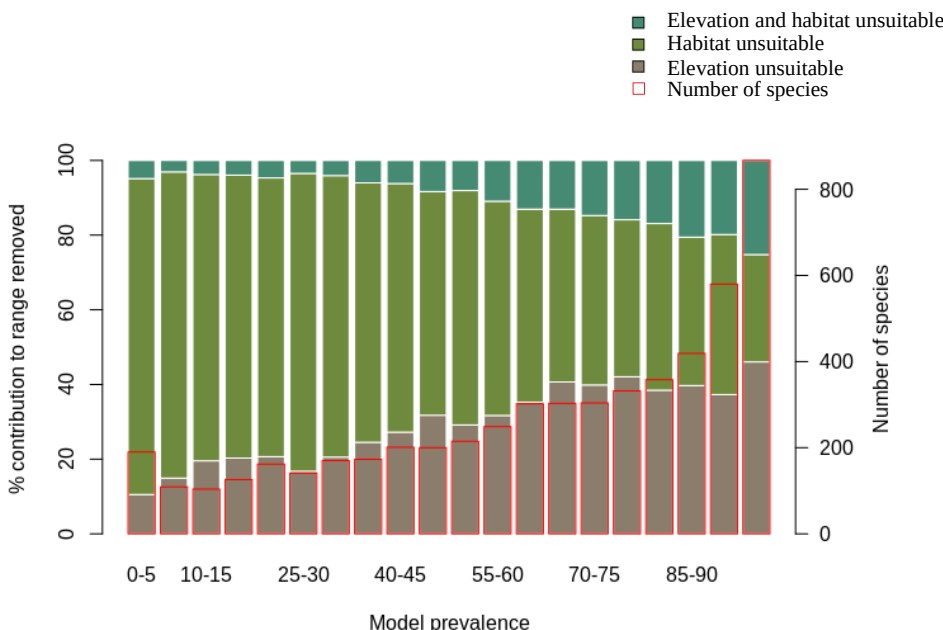

**Figure 2:** Percentage contribution of elevation range, habitat and both in clipping the IUCN range to AOH for mammals. See caption to Fig. 1 for interpretation.

For birds, the logistic model identified 178 AOH maps (1.7%) as lower outliers and 118 AOH maps (1.1%) as upper outliers out of 10475 AOH maps for 9163 terrestrial bird species. Similarly for mammals, the logistic model was applied to the AOH maps of 2758 species and identified 64 (2.3%) as lower outliers and 21 (0.8%) as upper outliers.

The mean of mid-point of elevation of the bird and mammal species identified as upper outliers was 2725 m and 3193 m respectively while the mid-point of elevation for species which were not identified as upper outliers was 1261 m for birds and 1289 m for mammals. This suggests that species identified as upper outliers were those found in higher elevation. These species were identified as upper outliers because the logistic models predicted low model prevalence at higher elevations. Also, the range maps for high-altitude species are drawn using contour maps, therefore most of the range is within the correct attitudinal band leading to high model prevalence for these species.

The lower outliers indicate where model prevalence was possibly underestimated due to potential errors in habitat mapping/coding and elevation range of the species. We found that the habitats





"Shrubland" and "Savannah" in the habitat-land cover crosswalk were not associated with the land
cover class "Herbaceous cover", leading to under-representation of these habitat types and hence
lower model prevalence than estimated by the logistic model (Fig. A1). We also found mismatch in
the elevation range and geographical range for the lower outliers (Fig. A2). There were few cases
where the range included large proportion of a particular land cover type which was not associated
with the suitable habitat of the species (Fig. A3). Moreover, we found that there was no land cover
information in the Copernicus land cover map for very small range polygons located on oceanic
islands which caused the AOH maps for these species to be empty. Furthermore, the land cover
class "open forest unknown" was discarded in the habitat land cover model. This led to low model
prevalence of AOH maps for some species whose ranges included this land cover. This was
corrected and a new set of AOH maps produced.
**Point validation**
Out of 4889 bird species (45% of all bird species) for which point data were available, 4689
(95.9%) had higher point prevalence than model prevalence and 200 species had lower point
prevalence than model prevalence (Fig. 3). The mean percentage of pixels coded as suitable inside
the 300 m buffers of point localities of 4889 species of birds was 62% (Fig. A5).



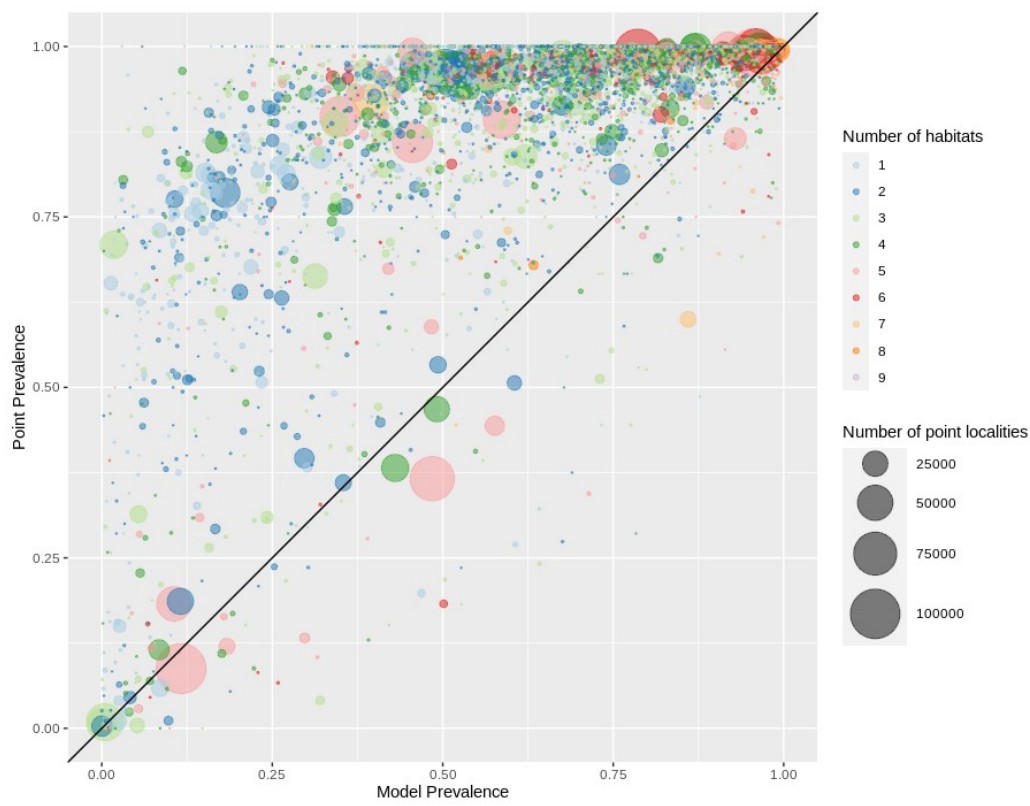

**Figure 3:** Point prevalence vs model prevalence for terrestrial birds. Colors indicate the number of habitats each species is coded to, size of circles indicates the number of point localities.

Out of 420 mammal species (8% of all mammal species) for which point data were available, 399 (95.0%) had point prevalence higher than model prevalence (Fig. 4). The mean percentage of pixels coded as suitable inside the 300 m buffers of point localities of 420 species of mammals was 78% (Fig. A5).



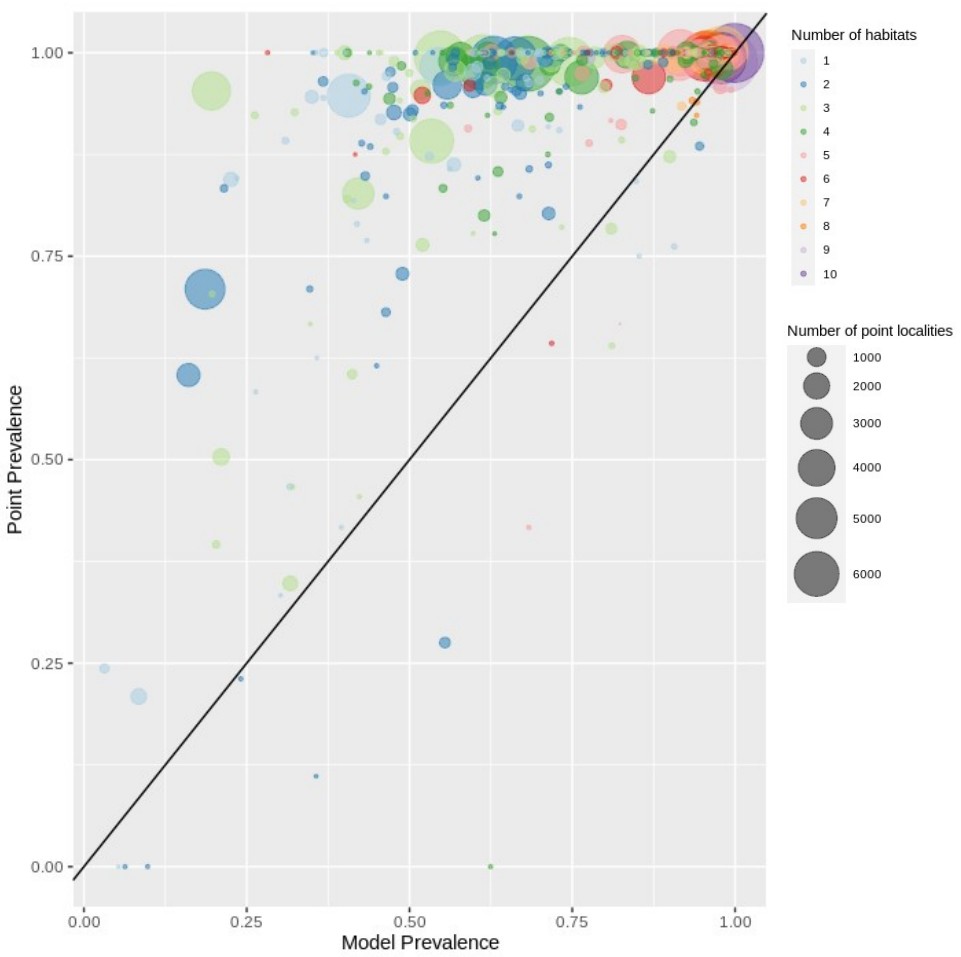

**Figure 4:** Point prevalence vs model prevalence for terrestrial mammals. Interpretation as in Fig. 3.

**Representativeness of validation sample**



We found that for birds over 60% all families, genera and orders were represented in the sample
included in the point validation and species from all biomes were represented but representation for
mammals was lower, as expected due to the much lower proportion of mammal species for which
point locality data were available (Fig. 5).
The validation points were spread across all of the variables and majority of their sub-classes (Fig.
A6, Fig. A7). Species with validation points tended to have larger range sizes, wider elevation

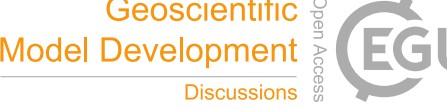

ranges and to be coded to more habitat classes than those without. Furthermore, validation points
were not available for any critically endangered or endangered mammals as these species are rare in
the wild.

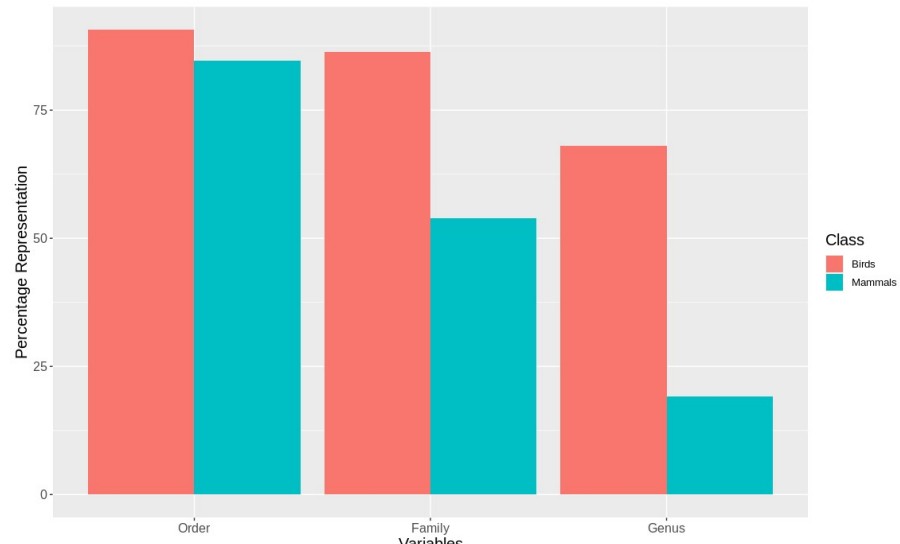

**Figure 5:** Taxonomic representativeness of validation sample for birds and mammals.
**Discussion**
On comparing our point validation results with previous validation analysis of AOH maps, we
found that validation results are similar to or better than previous exercises. For mammals,
Rondinini et al. (2011) evaluated AOH maps for 263 species at 300 m resolution, of which 241
(91.6 %) were better than random as compared to 95.0% in our analysis. However, it should be
noted that the mean model prevalence for AOH maps of Rondinini et al. (2011) was 54.8 ± 21.5 SD
as compared to 65.16 ± 25.42 for our AOH maps. The ratio of mean point prevalence to mean
model prevalence for Rondinini et al. (2011) was 1.4 compared to 1.38 in our case. Ficetola et al.
(2015) found that AOH for 94% of 115 amphibian species used in the validation analysis were
better than random with the mean model prevalence for species with validation points being 0.79 ±
0.21 SD. The ratio of mean point prevalence to mean model prevalence was 1.18 in this case.
Moreover, Catullo et al. (2008) found that 140 AOH maps out of 190 (73.7 %) South Asian
mammal species gave positive validation results while Rondinini et al. (2005) found the mean
proportion of suitable habitats correctly mapped inside the range for 181 species of African





vertebrates was 0.55 ± 0.01 SE using presence-absence data sets. The high validation success in our
analyses could be attributed to the use of novel habitat-land cover model (Lumbierres et al., 2021a),
the use of logistic regression models to identify systematic errors and the larger validation sample
as compared with previous exercises. Furthermore, the underlying land cover map used in
Lumbierres et al. (2021b), has the highest resolution among the global land cover maps providing
with more detailed land cover classification.
The point validation identified a small proportion of AOH maps which were no better than random.
Some of these had high model prevalence. In such cases, point prevalence must be exceptionally
high for the models to be better than random since even if a majority of point localities fall within
the AOH these maps may perform no better than random. For the AOH maps which were no better
than random and had low point prevalence, this was usually due to an apparent error in the coding
of elevation range of the species, the areas inside the range of the species where the point localities
fell being clipped out by what was assumed to be an erroneous elevation range. A list of species
with probably erroneous elevation coding will be forwarded to IUCN Red List team for future
corrections.
AOH maps aim to minimize the commission errors known to be present in species ranges without
increasing omission errors (Rondinini and Boitani, 2006). One of the limitations of this validation
analysis is the inability to quantify the commission errors of the AOH maps as we don't have the
true absence data of the species. Therefore, some uncertainty remains in AOH maps regarding the
commission errors.
Also, there are some intrinsic errors in the models as identified by the logistic regression analysis.
The species which are coded only to habitats like "Shrubland" might have under-represented model
prevalence as discussed above. However, the number of AOH maps identified as lower outliers by
the application of the logistic model was low for birds (178/10475) and for mammals (64/2758),
indicating that for the majority of AOH maps the observed model prevalence was fairly close to that
predicted by the model.



**Appendix A**

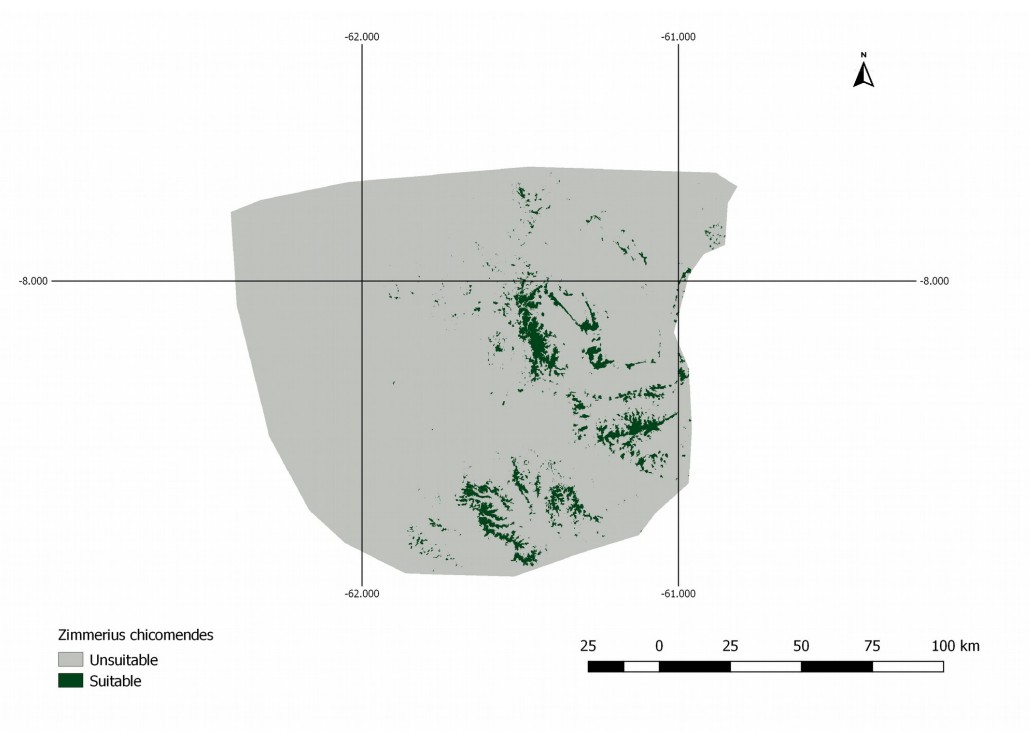

Zimmerius chicomendes
Unsuitable
Suitable

**Figure A1:** AOH map for species *Zimmerius chicomendesi*. The species is coded against "Forest"
and "Shrubland" habitats and the elevation range falls inside the IUCN range. However, the land
cover inside this range map includes a high proportion of "Herbaceous cover" land cover type
which is not associated with "Shrubland" habitat in the habitat – land cover association table.
Therefore, the model prevalence of this AOH is much lower than expected.





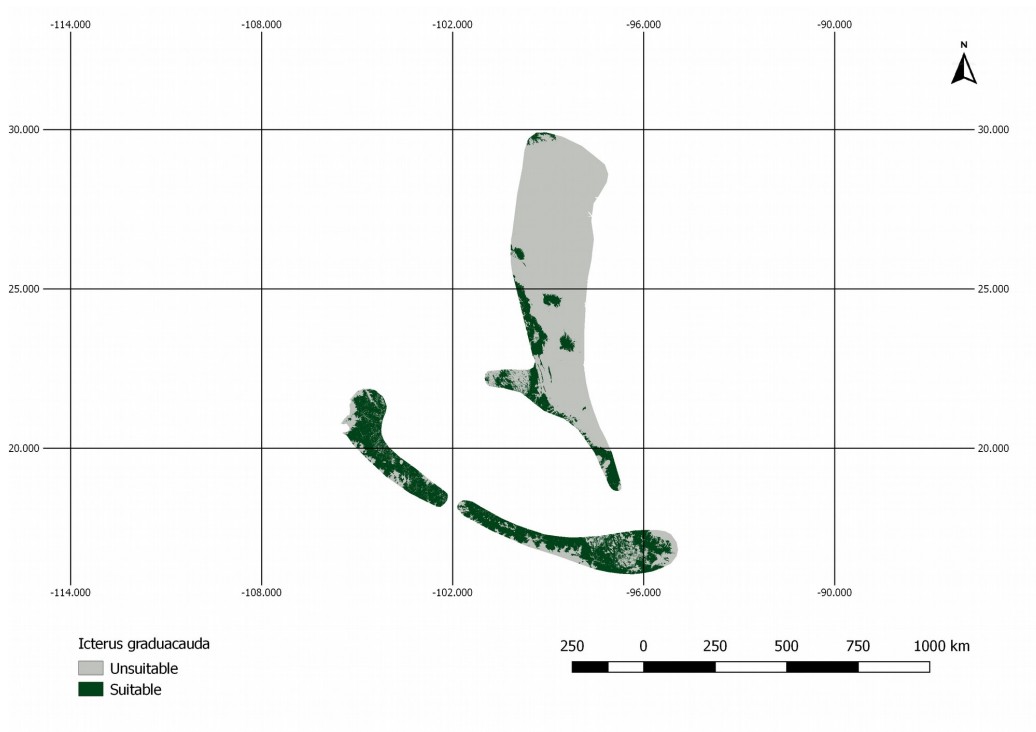

**Figure A2:** AOH map for the species *Icterus graduacauda*. The IUCN range of the species doesn't
cover much of the elevation range. Therefore, the model prevalence of this species is lower than
estimated.





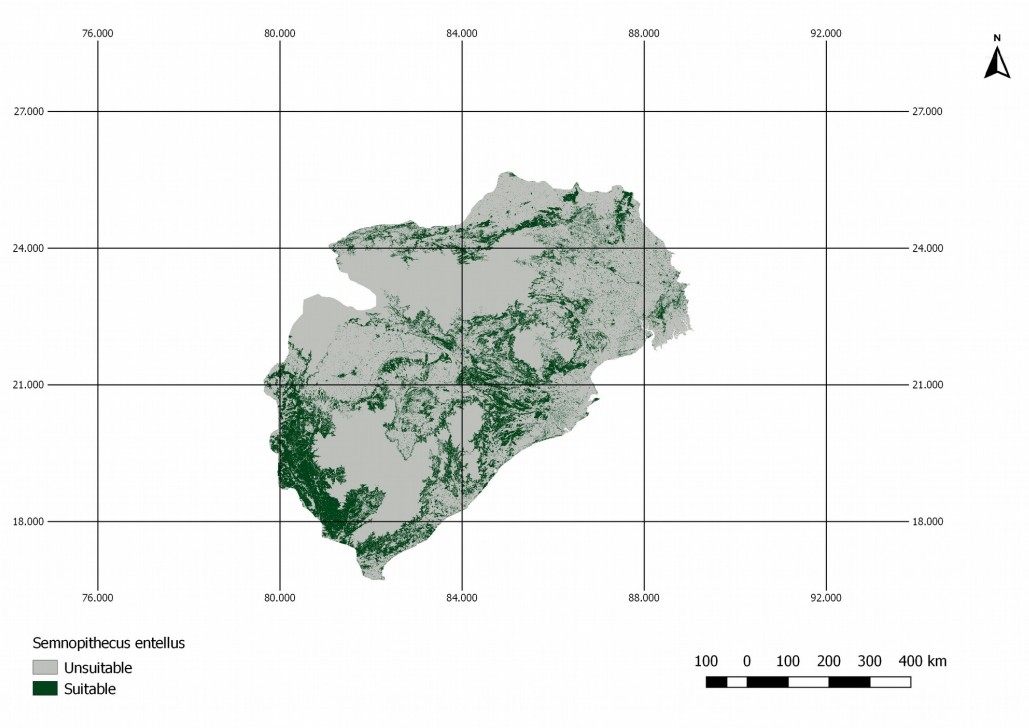

**Figure A3:** AOH for the species *Semnopithecus entellus*. There is a large proportion of land cover class "Cropland" inside the range map of this species. However, this species is not coded to habitats that are associated with the land cover "Cropland". Therefore, the model prevalence is lower than estimated.





361

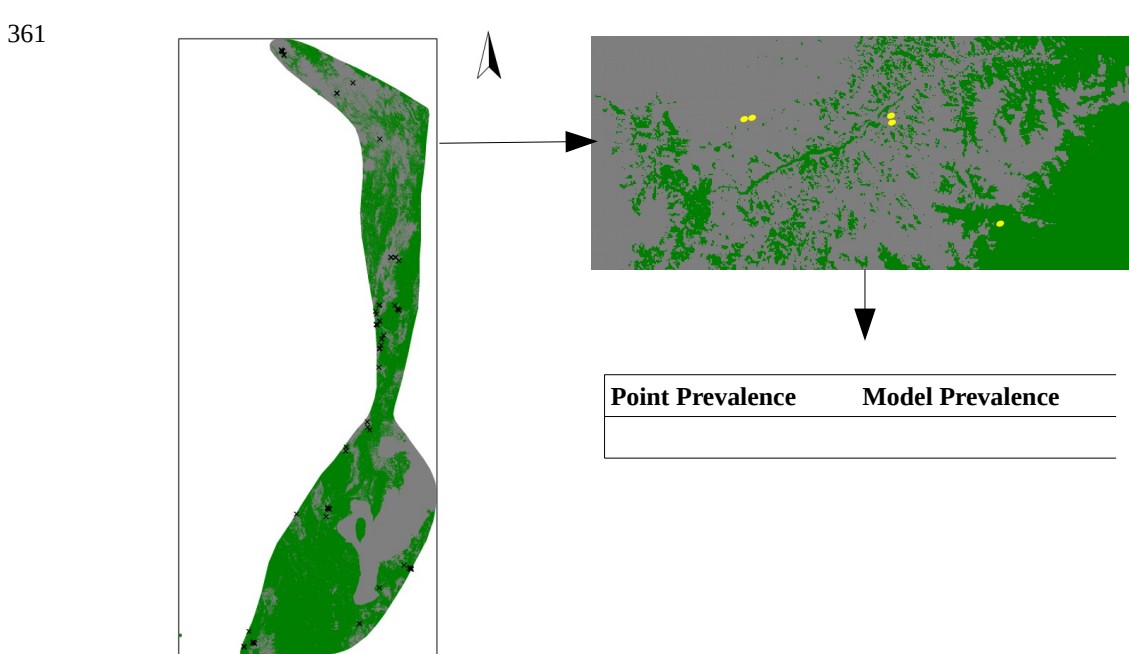

**Figure A4:** Point validation of the AOH maps using model and point prevalence.



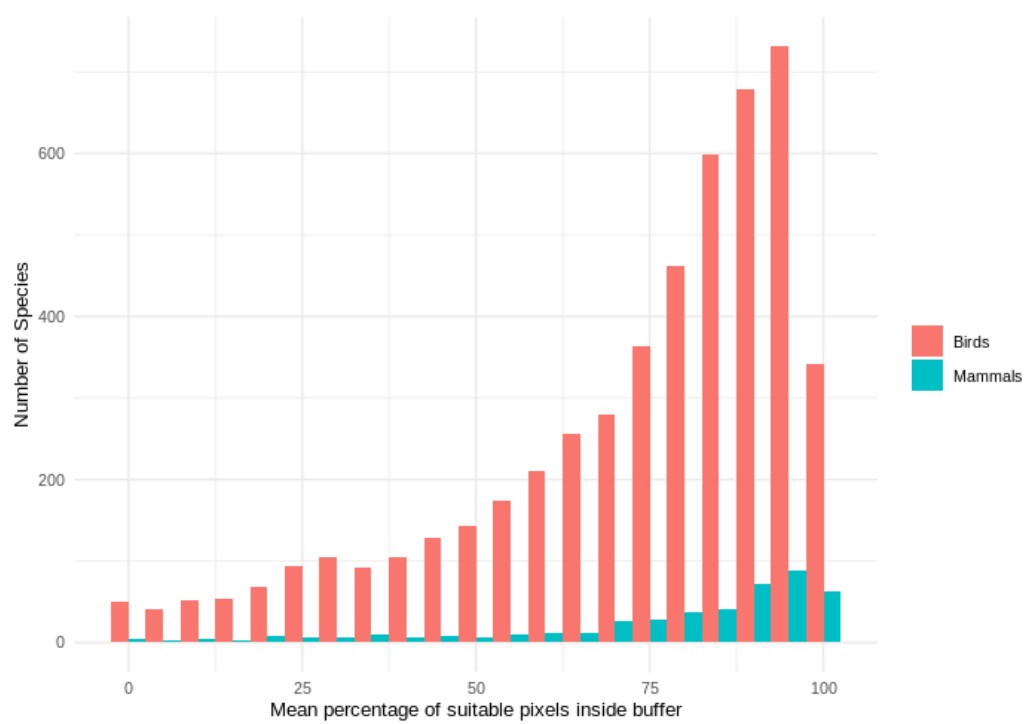

**Figure A5:** Histogram of mean percentage of suitable AOH pixels inside the 300 m buffer for mammals and birds species used in point validation.



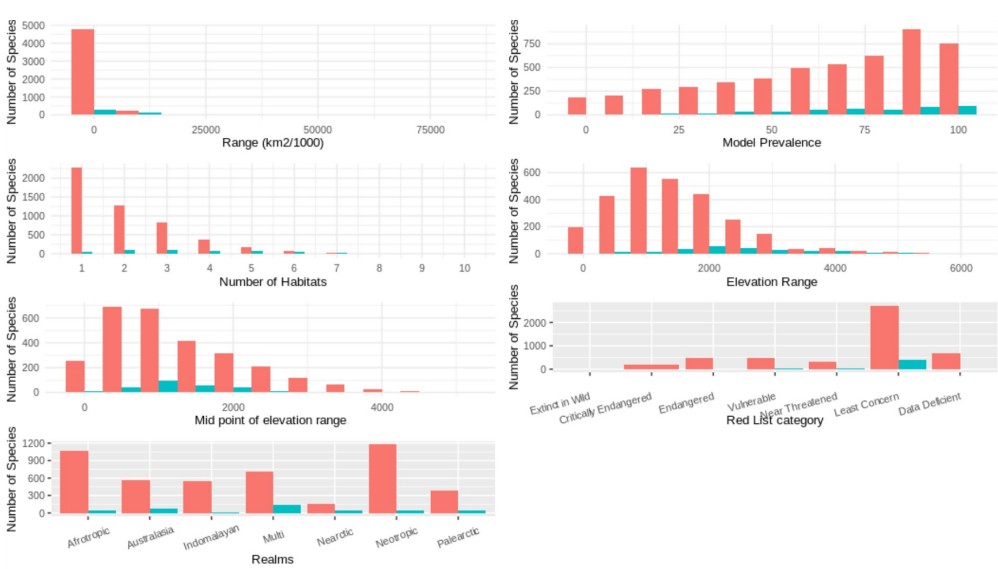

**Figure A6:** Comparison of species with and without validation points for mammals. Colours as in A5.

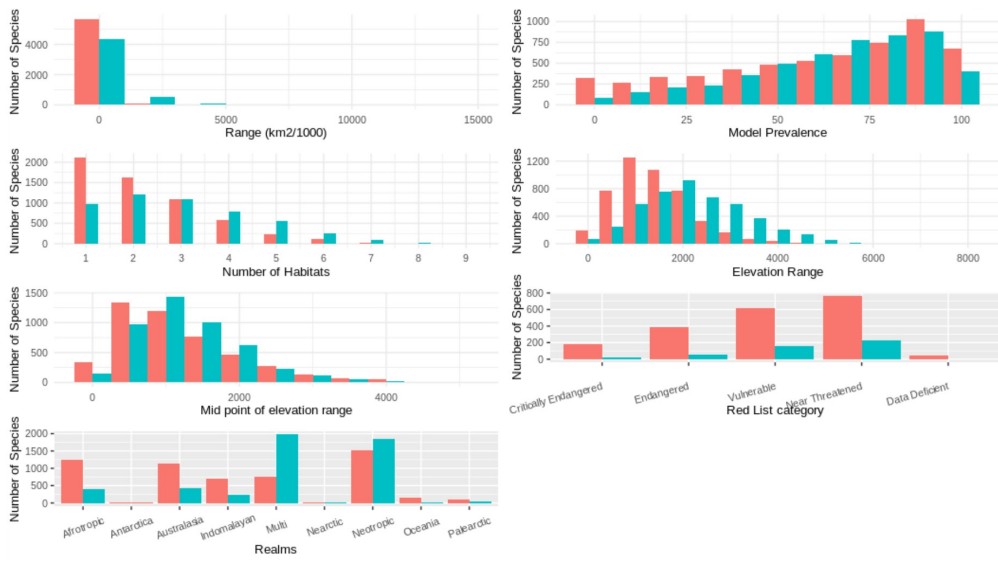

**Figure A7:** Comparison of species with and without validation points for birds. Colours as in A5.



**Data and code availability**

The point localities used in the validation analyses along with the metadata tables summarizing the validation analyses can be found at http://doi.org/10.5281/zenodo.5109073. The same DOI can be used to access the code used for validation and to also access some sample AOH maps which were validated.

**Author contribution**

PRD PFD and CR conceptualized the idea. PRD and ML curated and did the formal data analysis. PRD led the manuscript writing with contributions from all the authors. PFD CR SHMB supervised the whole process.

**Acknowledgment**

This research is part of the Inspire4Nature Innovative Training Network, funded by the European Union's Horizon 2020 research and innovation program under the Marie Skłodowska Curie grant agreement no. 766417.

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
