# Peer review of "A validation standard for Area of Habitat maps for terrestrial birds and mammals"

_Geoscientific Model Development, 2021_

## Author Response (AR1)

I would like to thank the reviewers for their valuable comments and also Matthew Rubino for the public comment. Following are my responses to the comments:

**Comment 1 from RC1:** Line 327 the work "with" is unnecessary in this sentence.

**Response:** We agree with the reviewer have removed "with" in the revised manuscript.

**Authors changes in the manuscript:** Removed "with" in page number 15, line number 428.

**Comment 2 from RC1:** Figure 3 and 4 - No critical but could be helpful - would it be possible to change the colors to a color ramp?

**Response:** We have changed the color ramp in the revised manuscript for figures 3 and 4.

**Authors changes in the manuscript:** Changed colors for Figure 3 (page 12) and Figure 4 (page 13).

**Comment 3 from RC1:** Figure A4 This figure requires more explanation or different graphics to better make the point.

**Response:** We have added more details to Figure A4 to add more clarity.

**Authors changes in the manuscript:** Figure A4 modified (page 19).

**Comment 4 from RC1:** Figure A6 & A7 – says colors are in A5 – in that legend the colors represent taxa, in this with or without validation points A6 Mammals – A7 Birds?

**Response:** We thank the reviewer for pointing out this error. We have corrected this by adding separate legends in Figures A6 and A7 where red represents species without validation points and green represents species with validation points.

**Authors changes in the manuscript:** Legends added for figures A6 and A7 (page 21).

**Comment from RC2:** I feel like the authors could be more practical and propositional on their interpretations along the discussion.

**Response:** We have added the following sentences in the discussion section to address this comment:

The AOH maps validated in this paper is the largest validation done till date in terms of number of species validated for birds and mammals. These maps will be freely available after the publication of Lumbierres et al. (2021b). We have also provided the metadata for the AOH maps of all the species along with validation statistics which can be used as a guideline by the users while using the AOH maps.

**Authors changes in the manuscript:** The above line was added at the end of discussion (page number 15, line number 449-453).

**Comment 1 from CC1:** Line 138: 10 point localities as a minimum seems low. Is there any reference for this quantity as an accepted threshold?

**Response:** Previous studies (Rondinini et al., 2011 and Ficetola et al., 2015) have used a minimum of 5 points with at least 1 point falling in a 1 km$^2$ grid. We have used a minimum of 10 points with at least 1 point falling in a 100 m$^2$ grid. Also, it must be noted that there is a trade-off between number of species included in the validation sample and the minimum number of points per species

required to be considered for point validation. As the threshold for minimum number of points per species increases, the number of species in the validation sample decreases. After filtering the point with three different filters, the number of points are already reduced and setting up the minimum number of points too high will reduce the validation sample greatly. With a threshold of 10 we have a good representative validation sample size.

**Authors changes in the manuscript:** No changes in the manuscript.

**Comment 2 from CC1:** Line 223: What does "one off pixel" mean?

**Response:** We mean one or few suitable habitat pixels falling inside the 300 m buffer during point validation. We have made the sentence simpler now by replacing "one off pixel" by "one or few pixels".

**Authors changes in the manuscript:** "one off pixel" removed in page number 8, line number 238 and replaced with "one or few pixels".

**Comment 3 from CC1:** Sentence spanning lines 226-228: What does this mean?

**Response:** We realized this sentence is complex and made it simpler now and it read as "It is therefore possible that the species included in the point validation analysis are not representative of the species not included."

**Authors changes in the manuscript:** Edited the sentence in page number 8, line number 241-242.

**Comment 4 from CC1:** Regarding "coordinate uncertainty": Did the author's include coordinate precision in their estimates of coordinate uncertainty?

**Response:** Yes we did consider the coordinate precision in our estimates of coordinate uncertainty. The points used in the validation have coordinate precision of 4 decimal places.

**Authors changes in the manuscript:** No changes in manuscript.

---

## Editor Decision (ED1)

**Editor comments on gmd-2021-245**

1) Please add a subheading at the beginning of section 2 to cover the part prior to present section 2.1.

2) Consider removing the background color in Figs. 3-5 and the last panels in Figs. A6+A7. For better readability and consistency consider fixing the plotting of bar pairs with missing data for species w/ or w/o validation points in Figs. A6+A7 (e.g. "Critically Endangered" in A6)

3) Confirm image credits and check color-blind friendliness as lined out in the editorial office's notification.